# Wireless theranostic smart contact lens for monitoring and control of intraocular pressure in glaucoma

Tae Yeon Kim[1], Jee Won Mok[2], Sang Hoon Hong [1], Sang Hoon Jeong[1], Hyunsik Choi[3], Sangbaie Shin[3], Choun-Ki Joo[2] & Sei Kwang Hahn [1,3] ✉

Glaucoma is one of the irreversible ocular diseases that can cause vision loss in some serious cases. Although Triggerfish has been commercialized for monitoring intraocular pressure in glaucoma, there is no smart contact lens to monitor intraocular pressure and take appropriate drug treatment in response to the intraocular pressure levels. Here, we report a precisely integrated theranostic smart contact lens with a sensitive gold hollow nanowire based intraocular pressure sensor, a flexible drug delivery system, wireless power and communication systems and an application specific integrated circuit chip for both monitoring and control of intraocular pressure in glaucoma. The gold hollow nanowire based intraocular pressure sensor shows high ocular strain sensitivity, chemical stability and biocompatibility. Furthermore, the flexible drug delivery system can be used for on-demand delivery of timolol for intraocular pressure control. Taken together, the intraocular pressure levels can be successfully monitored and controlled by the theranostic smart contact lens in glaucoma induced rabbits. This theranostic smart contact lens would be harnessed as a futuristic personal healthcare platform for glaucoma and other ocular diseases.

Glaucoma is a chronic eye disease requiring continuous medical care for patients' life time. Intraocular pressure (IOP) is one of the most important indicators of glaucoma[1,2]. Glaucoma patients are generally recommended for monitoring IOP and reducing IOP to a lower range. Accordingly, the appropriate IOP control is currently the only available first medical care for glaucoma patients. Commercial tonometers such as a rebound tonometer and a Goldmann applanation tonometer (GAT) have been used for the static IOP measurements in the clinic[3]. However, these measurements have some challenges. GAT can be inconvenient due to time and space constraints. Although home tonometry is portable, patients need to carefully follow the manual for accurate detection. These can significantly interrupt the daily life of patients. Furthermore, it is difficult to collect many data from the conventional static IOP measurements in a single day and to monitor

the IOP fluctuation of patients. This may cause inappropriate treatment to patients, because the IOP can change time to time[4,5]. Recently, eye drop solutions are the important first medical treatment for the IOP control of glaucoma patients[6–8]. However, it is highly challenging to achieve the high adherence of glaucoma patients. The method and time of drug treatments as instructed have not been followed by many glaucoma patients[9]. Furthermore, each patient can have a different therapeutic effect of drugs, which necessitate a different drug dose for each person[10]. Accordingly, it is important to monitor continuous IOP changes in real-time and delivers drugs in response to the IOP profiles.

Smart contact lenses can provide a new paradigm in glaucoma as a promising alternative to conventional management methods with non-invasive, continuous IOP monitoring and on-demand drug delivery[11,12]. In particular, recent smart contact lenses based on strain

[1]Department of Materials Science and Engineering, Pohang University of Science and Technology (POSTECH), 77 Cheongam-ro, Nam-gu, Pohang, Gyeongbuk 37673, Korea. [2]CK St. Mary's Eye Center, CK building, 559, Gangnam-daero, Seocho-gu, Seoul 06531, Korea. [3]PHI BIOMED Co., 168, Yeoksam-ro, Gangnam-gu, Seoul 06248, Korea. ✉e-mail: skhanb@postech.ac.kr

gauges, resonant, microfluidic, and optical systems have attracted great attention for continuous IOP monitoring including Triggerfish which achieved Food and Drug Administration approval in 2016 for the first time. Some of the smart IOP contact lenses have additional characteristics such as transparency and stretchability[13–15]. Furthermore, drug-eluting smart contact lenses increase the bioavailability of drugs in many folds compared to eye drops by prohibiting the washout of drugs with eye blinking and tears[16]. Smart contact lenses with drug complexes[17,18] and micelles[19] have been also developed for the treatment of glaucoma. However, these smart contact lenses still have some challenges for further applications such as low sensitivity, biocompatibility, and stability for long-term IOP monitoring without on-demand drug delivery functions[13–19].

Here, we report a highly integrated theranostic smart contact lens for both monitoring and control of IOP in glaucoma (Fig. 1a, b). The theranostic smart contact lens comprises of a gold hollow nanowire (AuHNW) based IOP sensor, a flexible drug delivery system (DDS), wireless circuits, and a ultra-low power application-specific integrated

circuit (ASIC) chip. For the IOP sensor, we synthesized AuHNW to achieve high sensitivity and stability with relatively high transparency for long-term IOP monitoring. Furthermore, the flexible DDS system with a biocompatible protective layer was fabricated for high drug-loading efficiency into a smart contact lens. All components were fabricated and highly integrated on-plane of parylene C substrate. The IOP sensor showed high sensitivity with high chemical stability and the flexible DDS could release timolol on-demand by the electrochemical dissolution of gold channels. We assessed the sensitivity of IOP sensor on the artificial eye and drug release efficiency in vitro. In vivo, we performed the demonstration of the theranostic smart contact lens for the successful monitoring and control of IOP in glaucoma animal models.

## Results

### Synthesis and characterization of AuHNWs

Hollow nanomaterials have many unique characteristics compared to bulk nanomaterials[20–24]. In particular, gold-based hollow nanomaterials

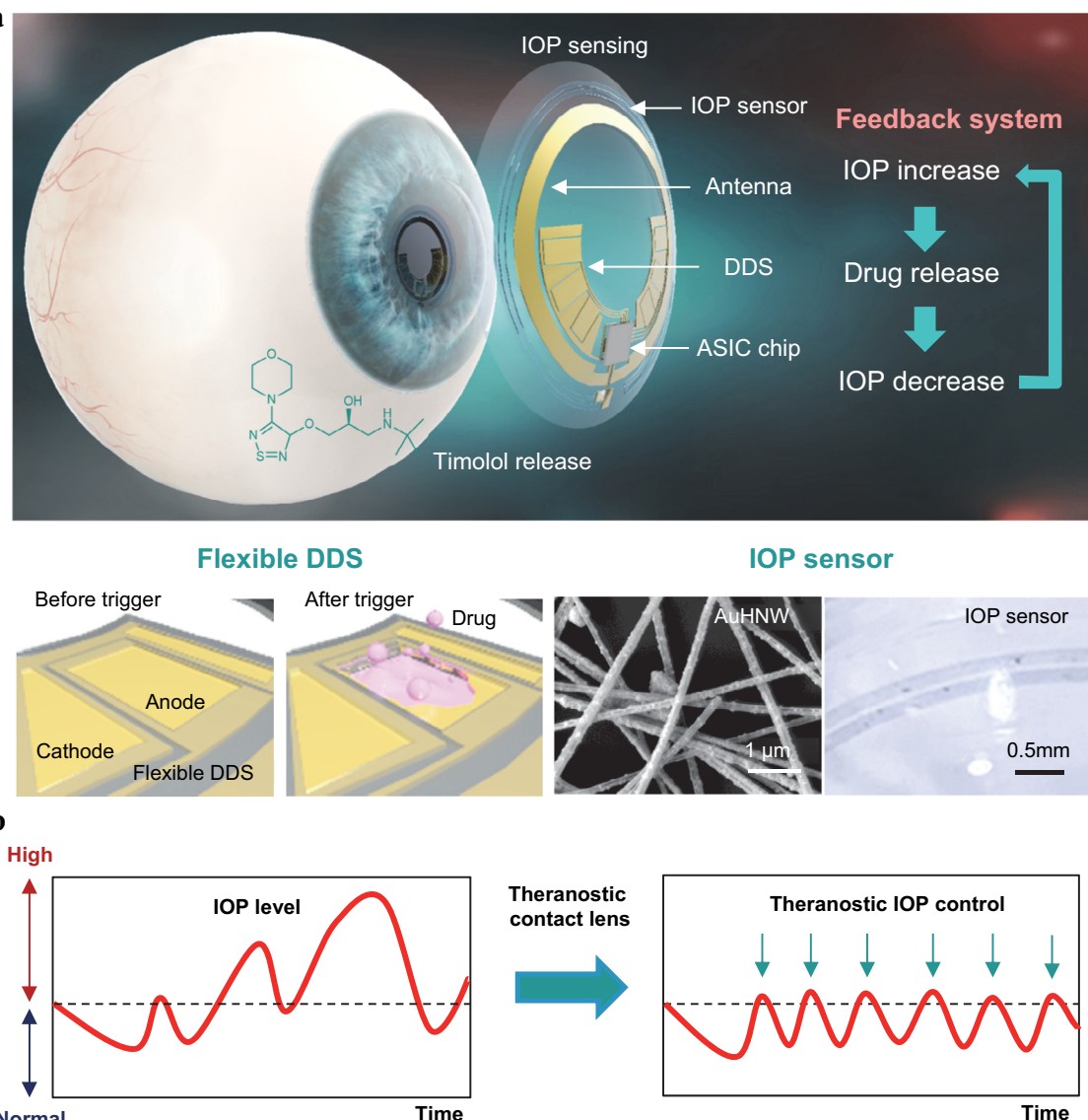

**Fig. 1 | Schematic illustration of a theranostic smart contact lens for glaucoma treatment. a** The structure of theranostic smart contact lens with a fully integrated AuHNW-based IOP senor, a DDS, and wireless circuits for wireless glaucoma treatments with a feedback system for IOP sensing and timolol release. **b** Schematic representation of the conventional continuous IOP monitoring and the IOP control by IOP monitoring and on-demand drug delivery for the treatment of glaucoma.

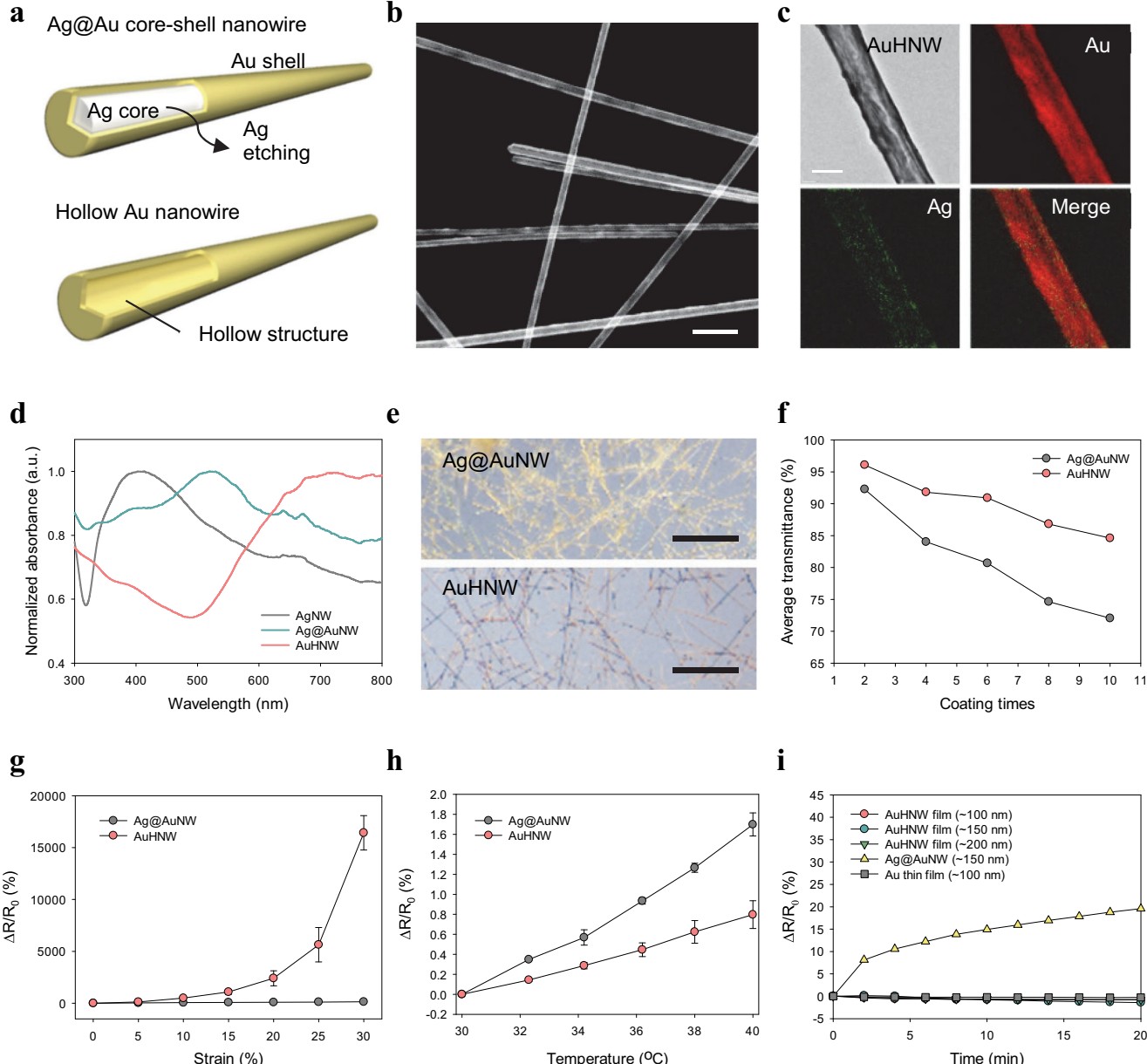

**Fig. 2 | Characteristics of synthesized AuHNWs. a** Schematic illustration of Ag@AuNW and AuHNW. **b** TEM images and **c** EELS images of AuHNWs (scale bar, 1 μm and 100 nm). **d** The normalized absorbance of AgNW, Ag@AuNW, and AuHNW. **e** OM images of Ag@AuNW and AuHNW films (scale bar, 5 μm). **f** The average transmittance change of each nanowire with increasing coating times. **g** Electromechanical (n = 6 independent experiments) and **h** thermoelectrical properties (*n* = 3 independent experiments) of Ag@AuNW and AuHNW. **i** The relative resistance change of nanowires and gold thin film under $H_2O_2$ (30%) exposure. **g**, **h** Data are presented as a mean value ± standard deviation (SD).

have been widely used for biomedical applications such as drug delivery[25], photothermal therapy[26], and surface-enhanced Raman scattering (SERS) sensors[27] with their unique optical properties and increased surface area. Here, AuHNW was synthesized by etching Ag core of Ag@Au core-shell nanowire (Ag@AuNW) for high transmittance, sensitivity, and chemical stability (Fig. 2a). The hollow structure of AuHNW (shell thickness of ca. 20–30 nm) was confirmed by high resolution−transmission electron microscopy (HRTEM) (Fig. 2b). Figure 2c shows the mapping for the distribution of Ag and Au to further confirm the hollow structure of AuHNW. The Ag core was selectively etched by diluted nitric acid, but the Au shell was not etched making the hollow structure. We directly compared the properties of AuHNW with those of Ag@AuNW. The absorbance peak was shifted into near-infrared (NIR) region as synthesized from AgNW to

AuHNW (Fig. 2d). Remarkably, the absorbance region of AuHNW was highly reduced in the visible region.

The optical microscopic (OM) images show the different optical properties in the visible region of Ag@AuNW and AuHNW (Fig. 2e). The nanowire solutions of AgNW, Ag@AuNW, and AuHNW showed different colors with their various optical properties (Supplementary Fig. 1). The average transmittance of AuHNW was high compared to that of Ag@AuNW in the overall visible region (Fig. 2f), which might be ascribed to that the absorbance peak was shifted into NIR region and reduced in the visible region. The dark field OM images showed two different coated nanowires and each area fraction was analyzed by the software program of Image J (Supplementary Fig. 2a). There were no critical differences in area fractions with two different coated nanowire films (Supplementary Fig. 2b). The transmittance spectra of

Ag@AuNW and AuHNW were well matched with our absorbance spectra as we assumed (Supplementary Fig. 2c, d).

We also investigated the electromechanical properties of AuHNW, which showed higher sensitivity than Ag@AuNW with a reasonable stretchability of 30% (Fig. 2g and Supplementary Fig. 3). In the scanning electron microscopy (SEM) images, AuHNW maintained the network structure like Ag@AuNW in the film states (Supplementary Fig. 4a). To investigate the effect of Au shell thickness on the electromechanical properties, we synthesized three AuHNWs with different shell conditions by changing the amount of AgNW templates. The shell thickness was decreased with increasing amount of AgNW (Supplementary Fig. 4b). The sensitivity and stretchability of strain sensors were changed according to the different shell thickness (Supplementary Fig. 4c). We optimized the amount of AgNW (50 mg) to synthesize AuHNW for a reasonable stretchability above 30% (Supplementary Fig. 4d).

For the characterization of thermoelectrical properties, AuHNW showed smaller resistance change than Ag@AuNW with increasing temperature (Fig. 2h). The temperature coefficient of resistance (TCR) of AuHNW was 0.08%/°C which was significantly lower than that of Ag@AuNW (0.17). We also investigated the chemical stability of each nanowire in physiological conditions by immersing into hydrogen peroxide ($H_2O_2$) solution (30% in water). Although the absorbance peaks of AgNW and Ag@AuNW were decreased by Ag etching with $H_2O_2$ after 2 h, the AuHNW showed no absorbance change (Supplementary Fig. 5a–c). The relative resistance was not significantly changed even in the presence of $H_2O_2$ (30% in water), which was similar to that of the gold thin film (Fig. 2i). Furthermore, there was no significant change in the relative resistance in PBS at 37 °C for 2 weeks (Supplementary Fig. 5d). These results confirmed the feasibility of AuHNW for long-term sensitive monitoring of IOP.

## Fabrication and Characterization of IOP sensor

The IOP smart contact lens sensor was fabricated in a circular design for monitoring the resistance change caused by the radial deformation of cornea with increasing IOP. The additional layer of poly(3,4-ethylenedioxythiophene):poly(styrenesulfonate) (PEDOT:PSS) incorporating D-sorbitol (D-PEDOT) was included to fill the empty space in micropatterned AuHNW networks for making a conducting path across individual nanowires. This hybrid structure could preserve the electrical conductivity in the micro-patterned structure. The IOP sensor was patterned by the conventional lift-off process after sequential coating of AuHNW and D-PEDOT (Fig. 3a). The sheet resistance of AuHNW increased 10 times higher than that of Ag@AuNW because of core etching and that of the hybrid sensor was slightly decreased because of the D-PEDOT additional layer (Supplementary Fig. 6a). After patterning the hybrid IOP sensor, thermoplastic polyurethane (TPU) was coated on the sensor for passivation, followed by transferring and embedding into smart contact lens. Although the transmittance of hybrid sensor was slightly decreased, it still had high transparency (above 84% at 550 nm; Supplementary Fig. 6b). As shown in Fig. 3b, the fabricated IOP sensor embedded into the smart contact lens was also highly transparent. Finally, the hybrid IOP sensor was successfully integrated with gold electrodes for wireless monitoring of IOP (Fig. 3c).

We optimized the thickness of parylene C and AuHNW coating to improve the sensitivity of IOP sensor. The thick parylene C substrate showed high elastic modulus that could not match with that of the cornea. It was not easily deformed with the curvature change of cornea. Although the thin parylene C coating increased the sensitivity of IOP sensor, the thin parylene C could not support the DDS structure[28]. The IOP sensor with thin parylene C showed much higher sensitivity (300 nm and 32% at 35 mmHg) than that of thick parylene C (1500 nm and <1% at 35 mmHg) (Fig. 3d). AuHNW was spin-coated by five times to form the electrical path and the IOP sensor was patterned in double

lines for the successful fabrication in the limited area of smart contact lens. The sensitivity was increased 11 to 32% at 35 mmHg by optimizing coating times and the sensor design. There was high correlation between IOP and relative resistance changes (Fig. 3e). We could also monitor real-time IOP as small as 3 mmHg (Supplementary Fig. 7a) and the repeated IOP fluctuation without any sensitivity change less than -0.07% (Supplementary Fig. 7b). As shown in Supplementary Fig. 7c, there was no critical hysteresis when the IOP was recovered after increasing to 35 mmHg (<-0.14%). Finally, IOP levels were measured repeatedly from 0 to 35 mmHg to confirm the reliability of sensor. There was no significant difference in the profile of relative resistance change according to IOP changes (Fig. 3f). Furthermore, we assessed the IOP sensors containing different nanowires of AuHNW and Ag@AuNW (a control) for comparison (Supplementary Fig. 7d, e). The IOP sensor with AuHNW showed much higher sensitivity (over 75% at 200 mmHg) than that of Ag@AuNW (less than 5% at 200 mmHg). We confirmed that the AuHNW-based IOP sensor could be successfully operated even in the higher IOP range with high sensitivity.

## Fabrication and characterization of flexible DDS

We designed two different types of DDS for daily and weekly uses (Fig. 4a, b). The flexible DDS was fabricated on the parylene C substrate by depositing gold thin film (thickness of 100 nm) and integrating with the wireless circuits (thickness of 500 nm). The drug reservoirs were fabricated by photolithography of SU-8 with a thickness of 50 µm and loaded with the IOP-lowering drug powder of timolol. After that, the biodegradable polymer, poly(vinyl alcohol) (PVA), was coated on the loaded drugs as a protection layer. Finally, the additional parylene C layer (100 nm) was deposited on the DDS for drug sealing. The fabricated DDS was transferred on the supporting substrate and the back side of parylene C was selectively etched by reactive ion etching (RIE) to open the gold channel of DDS. The opened gold channel could be selectively dissolved in the phosphate-buffered saline (PBS) by applying an electrical voltage of 1.85 V. As shown in Fig. 4c, d, the opened gold channel was fully dissolved by the electrochemical reaction between gold and $Cl^-$ in the PBS or physiological fluid according to the following equations: $Au + 4H_2O \leftrightarrow Au(H_2O)_4^{3+} + 3e^-$ and $Au(H_2O)_4^{3+} + 4Cl^- \leftrightarrow AuCl_4^- + 4H_2O$[29]. The electrochemical reaction between gold and $Cl^-$ was analyzed by measuring the current-time ($I$–$t$) curve during the gold dissolution at the constant voltage of 1.85 V. As shown in Supplementary Fig. 8a, gold channels were fully dissolved within 160 s and the operation current was in the range from 5 to 9 µA. Furthermore, the gold channels could be dissolved in the artificial tear in the same manner with that in PBS (Supplementary Fig. 8b). The mechanical bending tests confirmed the flexibility of DDS to be embedded into a smart contact lens. As shown in the $I$–$t$ curve, the dissolution time of bended DDS was within 140 s and the operation current was ca. 6 µA at the curvature of 6.25 mm (Fig. 4e). Because the curvature of contact lens is generally ca. 8 mm, we confirmed that this DDS was flexible enough to be embedded into the smart contact lens without any critical damages[30,31].

In order for therapeutic applications, it is important to load a sufficient amount of drug in the limited area of small reservoirs in a smart contact lens[16,29,32]. The drugs were loaded in powder form to increase the amount of drug loading. The sufficient drugs could be loaded in six reservoirs with 42.78 µg for a daily type and 218.28 µg for a weekly type (Supplementary Fig. 8c). The amount of loaded drug in each type corresponded to that in the one drop of commercial eye drop (50 µg). The parylene C as a sealing material penetrated into the powder during chemical vapor deposition and made difficult the drug dissolution. To solve this problem, biodegradable PVA polymer was introduced as a protection layer. As shown in Fig. 4f, while the timolol release was lower than 50% without the PVA layer, the drug was fully dissolved out in the presence of PVA. Although the released timolol was <5% without electrical triggering for 90 min, ~85% of the loaded

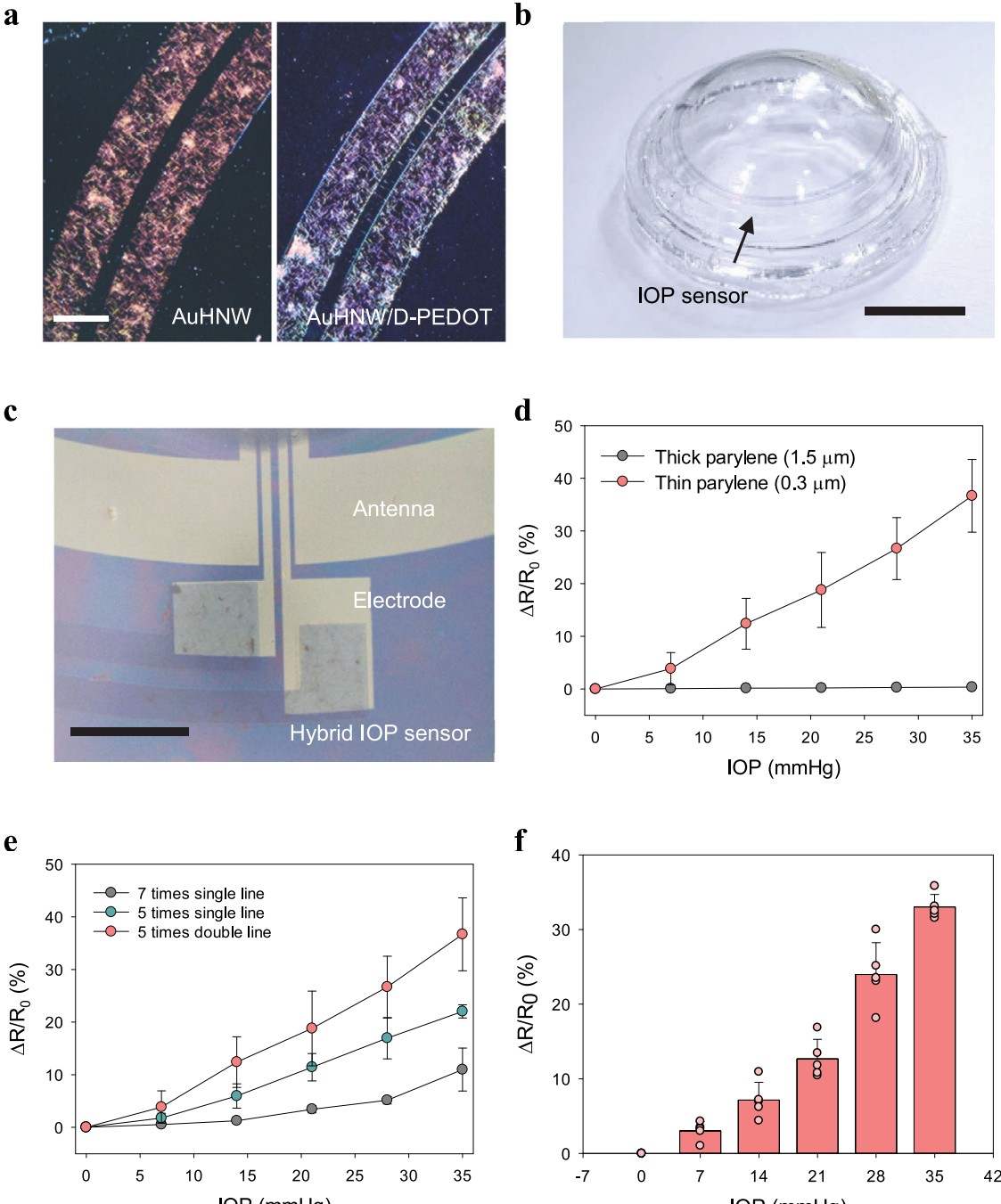

**Fig. 3 | Characteristics of IOP sensor. a** OM images of patterned IOP sensors (scale bar, 200 μm). **b** Photo-image of an IOP sensor embedded into a smart contact lens (scale bar, 6 mm). **c** OM image of an integrated IOP sensor and wireless circuits (scale bar, 600 μm). The relative resistance change of IOP sensors with **d** different substrate thicknesses ($n = 3$ independent experiments) and **e** coating times and designs ($n = 3$ independent experiments). **f** Five different measurements of relative resistance with increasing IOP by using a single IOP sensor ($n = 5$ independent experiments). **d**–**f** Data are presented as a mean value ± SD.

timol was released out within 5 min and almost all the loaded timolol was fully released out within 30 min (Fig. 4g). We investigated the cumulative release of timolol by sequentially activating three different reservoirs one by one at 10, 25, and 40 min with 15 min intervals. As shown in Fig. 4h, there was no significant release of timolol until 10 min, but the following release of timolol was dramatically enhanced at each activation time of DDS.

**Biosafety of theranostic smart contact lens**

A variety of smart contact lenses have been widely investigated with diverse nanomaterials, but the biosafety issues of their nanomaterials have not been fully addressed[33–36]. Especially, silver-based nanomaterials for transparent or stretchable contact lens might damage cells or DNAs by releasing $Ag^+$ ions[37]. We conducted live/dead fluorescent staining to assess the biosafety of nanomaterials in the IOP sensor. The cells were directly seeded on the parylene C substrate as a blank control, PEDOS:PSS films incorporating D-sorbitol (D-PEDOT) and AuHNW films for 3 days in an oven at 37 °C. As shown in Fig. 5a, live cells were stained in green and dead cells were in red on the fluorescent optical microscope images. Although most of cells were dead in the case of silver nanowire by the released $Ag^+$ ion (<15%), >92% of cells survived in all kinds of materials in the IOP sensor (Fig. 5b).

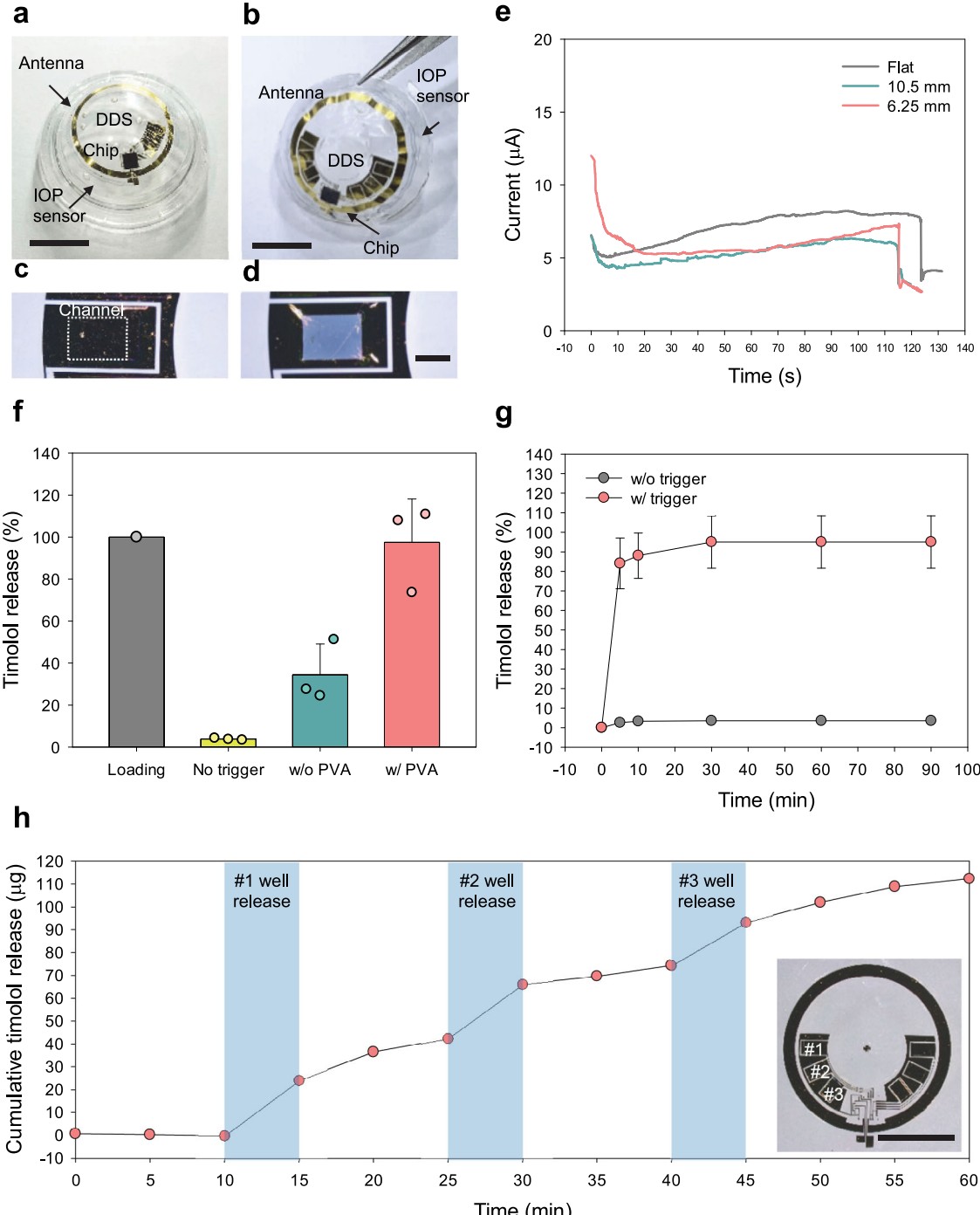

**Fig. 4 | Characteristics of flexible DDS.** Optical images of **a** daily and **b** weekly types of theranostic smart contact lenses (scale bar, 5.5 mm). **c** Before and **d** after the selective electrochemical dissolution of the gold channel in PBS (scale bar, 400 μm). **e** I–t curve of flexible DDS with different bending conditions. **f** In vitro release of timolol from the flexible DDS (*n* = 3 independent experiments). **g** In vitro release profile from a reservoir (*n* = 3 independent experiments) and **h** the cumulative release profile from three different reservoirs (scale bar, 5.5 mm). **f, g** Data are presented as a mean value ± SD.

Furthermore, we assessed the temperature change of smart contact lens to check the thermal biosafety. Finally, we confirmed no corneal damage by wearing smart contact lens (Fig. 5c). There was no inflammation in the cornea (Fig. 5d) and the thickness of cornea was not significantly changed by wearing the smart contact lens (Fig. 5e). As shown in Fig. 5f and Supplementary Fig. 9, infrared thermal camera analyses showed no significant temperature increase in the smart contact lens on the rabbit eye during IOP data collection and DDS activation by wireless communication and electrical signaling. All these

results revealed the biosafety of our theranostic smart contact lens without critical damage in the glaucoma-induced animal models.

## In vivo IOP monitoring and control of theranostic smart contact lens

The IOP sensor, DDS and wireless circuits, and ASIC chip were integrated and embedded into a smart contact lens. Figure 6a shows the fully integrated theranostic smart contact lens on the rabbit eye. For the IOP sensing, smart contact lens was designed with a diameter of

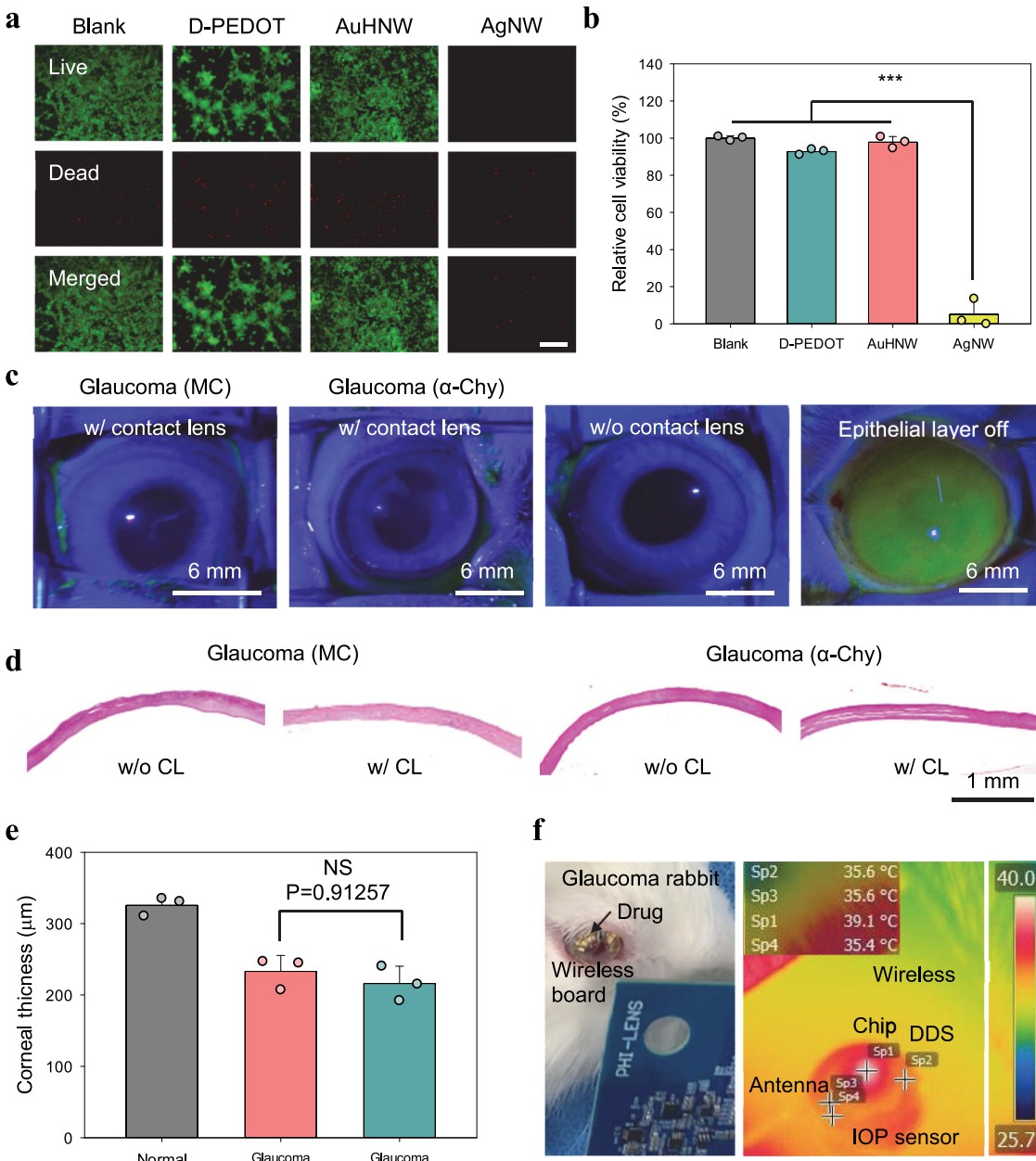

**Fig. 5 | Biosafety assessment of nanomaterials and smart contact lens.**
**a** Fluorescent microscopic images of NIH 3T3 cells after the live/dead assay (scale bar, 200 μm). **b** The relative cell viability of each sample ($n = 3$ independent experiments, ***$P = 0.00003$ for Blank and AgNW, ***$P = 0.00004$ for D-PEDOT and AgNW, ***$P = 0.00004$ for AuHNW and AgNW). **c** Fluorescein staining for corneal damage in rabbit eyes. **d** OM images and **e** the corneal thickness analysis of

glaucoma-induced rabbits' corneas ($n = 3$ independent experiments). **f** Photo-images for wireless power transmission and communication, and the thermal characterization of theranostic smart contact lens on rabbit's eye with a control board. **b, e** Data are presented as a mean value ± SD. One-sided statistical analyses were performed by using a one-way analysis of variance.

7.5 mm for perfect fitting on the rabbit eye (Supplementary Fig. 10). To assess the IOP control of theranostic smart contact lens, glaucoma was induced by injecting methylcellulose (MC) into anterior chamber or α-chymotrypsin (α-chy) into posterior chamber of rabbits. The glaucoma rabbits showed high IOP levels (over 25 mmHg) compared to normal rabbits. (Supplementary Fig. 11). After that, we assessed the theranostic smart contact lens for the IOP control by simultaneous IOP monitoring and on-demand drug delivery (Supplementary Fig. 12a). The rabbits were fixed in the cages and the theranostic smart contact lenses were worn on the rabbit eyes (Supplementary Fig. 12b, c). We previously reported the detailed principle of our wireless communication systems[38]. The overall architectures of ASIC chip and wireless board are shown in Supplementary Fig. 13. The change of output codes was

correlated with the IOP of each rabbit with a different eye size (Supplementary Fig. 14). The recorded output code was 0 mmHg before wearing smart contact lens and the output code was measured at more than two different IOP levels after wearing smart contact lens. The code change (%) by smart contact lens showed high correlation ($R^2$, >0.98) with the IOP (mmHg) by a tonometer. After calibration for this correlation, the measured IOP (mmHg) by the smart contact lens was compared with IOP (mmHg) by the tonometer. As shown in Fig. 6b, there was a strong correlation between the measured IOP levels by the tonometer and the smart contact lens with a coefficient of determination ($R^2$) of 0.94, indicating good intersession reproducibility. Furthermore, Bland-Altman analysis was performed to confirm the agreement of IOP measured by the smart contact lens and the

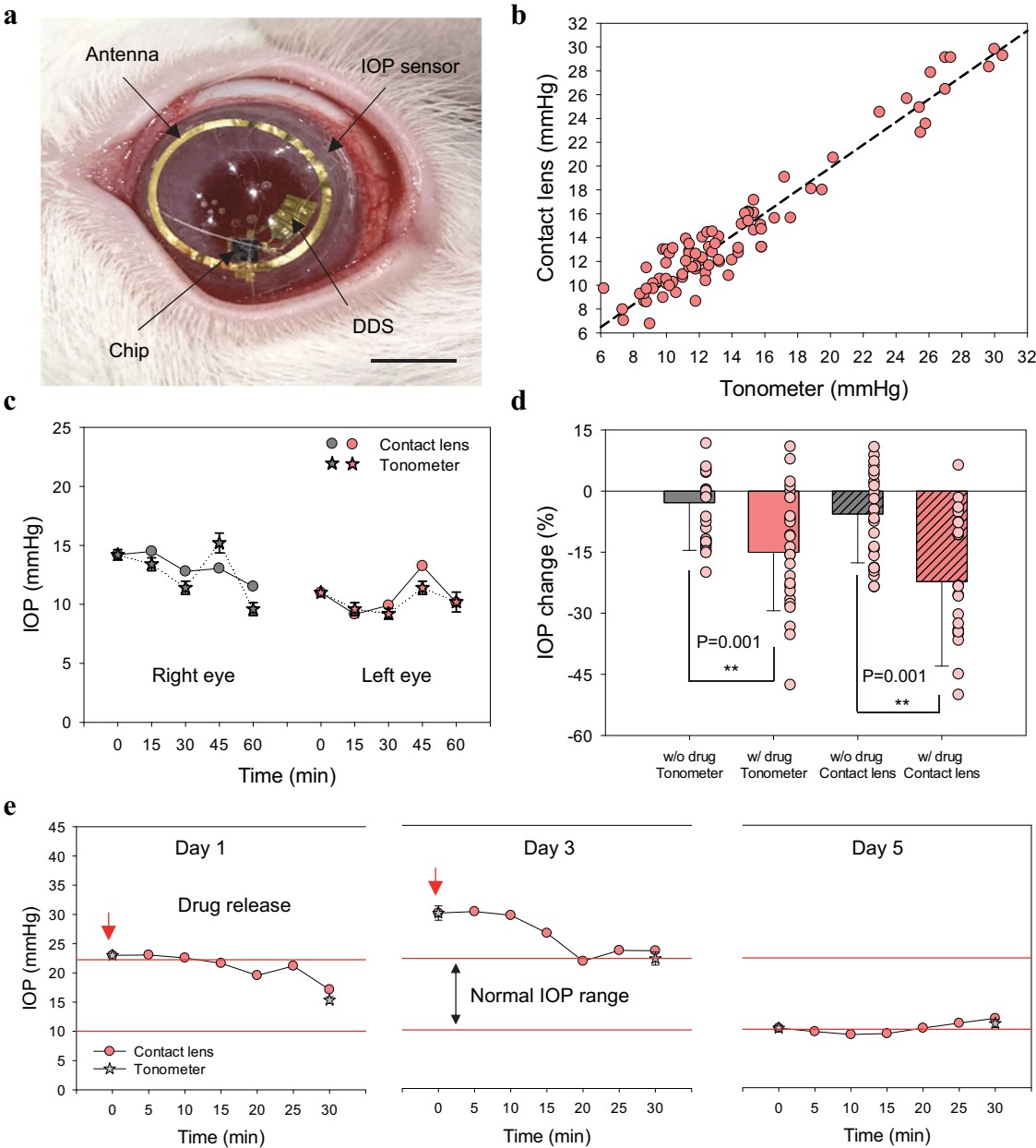

**Fig. 6 | In vivo IOP control with a theranostic smart contact lens. a** Photo-image of a fully integrated theranostic smart contact on rabbit's eye (scale bar, 5.5 mm). **b** The correlation of IOP measured with a smart contact lens and a tonometer ($R^2 = 0.94$). **c** IOP monitoring of right and left eyes with a smart contact lens and a tonometer ($n = 3$ independent experiments for tonometer). **d** The IOP change by timolol released from the smart contact lens ($n = 25$ independent experiments,

$^{**}P < 0.01$). For statistically analysis, one-sided statistical analyses were performed by using one-way analysis of variance. **e** The IOP control of the theranostic smart contact lens by simultaneous IOP monitoring and timolol release ($n = 5$ independent experiments for tonometer except for Day 1 ($n = 3$)). For **c**–**e** data are presented as a mean value ± SD.

tonometer (Supplementary Fig. 15). The limit of agreement (LoA) between the smart contact lens and the tonometer was in the range from −3.16 to 2.90 mmHg with 95% confidence interval. We monitored the IOP profiles of oculus dextrus (OD) and oculus sinister (OS) eyes of a rabbit for 60 min at 15 min intervals with the smart contact lens and the tonometer (Fig. 6c). The IOP profiles showed a similar trend in each eye.

Figure 6d shows the IOP change with and without the release of timolol from the smart contact lens. The released timolol could reduce ca. 22% of IOP compared with non-treatment cases. Furthermore, there was no significant difference in the IOP change measured by a smart contact lens and a tonometer regardless of the timolol release. Remarkably, upon the drug treatment with smart

contact lens, the IOP was decreased quickly, maintained within the reduced IOP for 18 h, and returned to the initial IOP level after 24 h (Supplementary Fig. 16a). Finally, the IOP level was controlled with the theranostic smart contact lens by monitoring IOP and releasing timolol every other day for 5 days (Fig. 6e and Supplementary Fig. 16b). On the first day, because the IOP level was in the high IOP range (above 22 mmHg), timolol was released out to reduce IOP and the IOP was reduced below the normal range. On the third day, IOP was still above the normal range and the released timolol could reduce the IOP level near the normal range. On the final day, the IOP level was in the normal range and timolol was not released out from the smart contact lens. All these results confirmed the feasibility of our fully integrated theranostic smart contact lens for monitoring

and control of the IOP with the feedback system of an IOP sensor and a DDS for further futuristic glaucoma treatment.

### In vivo therapeutic effect by theranostic smart contact lens

The therapeutic effect of timolol released from the theranostic smart contact lens was analyzed and compared with that of eye drop (0.5% Timoptic) in glaucoma-induced rabbits. The right eyes of glaucoma rabbits were treated by eye drop every day or timolol released from the smart contact lens every other day for 2 weeks (Supplementary Fig. 17). The left eyes of glaucoma rabbits were treated with PBS as a negative control. As shown in Supplementary Figs. 18a and 19a, retina histology revealed that the retina thickness was similar with that of the normal in both treatment groups. However, the thickness of retina was thin in the case of non-treatment groups. Especially, there were significant differences in ganglion cell layers (GCL) with and without treatment. Furthermore, the inner nuclear layers (INL) of non-treatment groups were unstable and thinned compared to those of normal and treatment groups.

We also carried out immunohistochemical analyses to further confirm the therapeutic effect of timolol released from the smart contact lenses. Immunohistochemically stained images showed significant differences between treatment and non-treatment groups of glaucoma rabbits. The retinal gliosis marker of glial fibrillary acidic protein (GFAP) related to retina damages was increased in the case of a non-treatment group, but the GFAP expression of treatment groups was similar to that of normal groups (Supplementary Figs. 18b and 19b). Furthermore, CD11b related to optic nerve injury was mainly expressed in glaucoma retina and non-treatment groups (Supplementary Figs. 18c and 19c). The brain-derived neurotrophic factor (BDNF) and the brain-specific hmeobox/POU domain protein 3 A (Brn3a) which were related to the retinal ganglion cell marker were highly expressed in the normal and treatment groups (Supplementary Figs. 18d, e and 19d, e). Glaucoma is a chronic eye disease with progressive neuropathy characterized by the death of retinal ganglion cells and the thinning of nerve fiber layers. Taken together, we could confirm the feasibility of theranostic smart contact lens for prohibiting the progression of glaucoma[39].

## Discussion

Smart contact lenses have been developed using a variety of nanomaterials for diagnostic and therapeutic applications. Especially, smart IOP contact lenses were developed with various nanomaterials of graphene[11], AgNW[14], and silicon[12]. In this work, we used AuHNWs with some unique optical and electromechanical properties, and biocompatibility. For comparison, Ag@AuNW was used as a control with no significant difference in the morphology from that of AuHNW (Supplementary Fig. 20). First, the absorbance region of AuHNW can be shifted into NIR region because of their surface plasmon resonance effect, resulting in high transparency in the visible region. Furthermore, AuHNW with a thin shell can be easily fractured under the applied strain compared to bulk nanowires, showing high sensitivity and reasonable stretchability (Supplementary Fig. 21). Although some sensor materials should be passivated due to the biocompatibility and stability issues, AuHNW is highly biocompatible and stable under the physiological condition even without any passivation for further clinical development and applications.

Although previous smart IOP contact lenses could continuously monitor IOP, there has been no report on the control of IOP in response to the IOP monitoring. We confirmed that our theranostic smart contact lens could reduce the IOP of glaucoma-induced rabbits in different IOP levels (Supplementary Fig. 22). The progression of glaucoma was successfully prohibited with the well-preserved structure of retina. A sufficient amount of drugs could be loaded into the flexible DDS and released on-demand by electrical triggering for the control of IOP. Because each glaucoma patient has a different

glaucoma progression, target IOP is generally different for each glaucoma patient. In addition, the side effects and IOP reduction efficiency are different for each glaucoma patient with different patient adherence. Accordingly, it is highly important to have the optimal therapeutic plan for each glaucoma patient. Our theranostic smart contact lens to continuously monitor IOP and deliver the necessary drugs would maximize the therapeutic effect and minimize the side effect with a personalized treatment plan for futuristic glaucoma treatment. This futuristic smart contact lens for personalized IOP control would open a new paradigm in the field of glaucoma.

In summary, we have successfully developed a highly integrated smart theranostic contact lens containing a sensitive AuHNW-based IOP sensor, a flexible DDS, an ASIC chip, and wireless power and communication systems for both monitoring and control of IOP in glaucoma. A flexible material of AuHNW was synthesized to fabricate a highly sensitive, stable, and biosafe IOP sensor for long-term IOP monitoring. The IOP-lowering drug timolol was loaded into the flexible DDS with a biocompatible protection layer, which could be released from the reservoir by electrical triggering for the control of IOP. The biosafety of smart theranostic contact lenses was confirmed by live/dead assay of nanomaterials, and the analyses of corneal damage, corneal thickness change, and thermal damage. Furthermore, in vivo experiments in glaucoma-induced rabbits confirmed the feasibility of theranostic smart contact lens for both monitoring and control of IOP. Finally, the therapeutic effect of theranostic smart contact lens was successfully demonstrated by the analysis of the retinal structure and four different biomarkers of GFAP, CD11b, BDNF, and Brn3a. This theranostic smart contact lens with a feedback system would open a new avenue to personalized glaucoma treatment (Supplementary Table 1).

## Methods

### In vivo study approval

In vivo tests were performed with male New Zealand White rabbits weighing 1.7–2.5 kg with no sign of ocular inflammation. All in vivo experiments were conducted in accordance with the ARVO statement for the use of animals in ophthalmic and vision research with the approval by the institutional care and use committee (CRONEX-IACUC: 202110005, CRONEX, Korea).

### Synthesis and characterization of AuHNWs

AuHNWs were synthesized by selective etching of Ag core in 30% of dilute nitric acid (SAMCHUN). The nitric acid was slowly added into the Ag@AuNW solution at a volume ratio of 1:1. After 1 h, AuHNWs were washed with ethanol and DI water multiple times for further characterization. The structure of AuHNW was investigated by electron energy loss spectroscopy (EELS) equipped with HRTEM (JEM-2200FS, JEOL). The networks and structures of AuHNW and Ag@AuNW were analyzed by SEM (MIRA3, TESCAN).

### Optical characterization

For the characterization of optical transmittance, AuHNW and Ag@AuNW were spin-coated on a parylene C (LAVIDA 110, Femto Science) substrate after plasma treatment. For the characterization of absorbance, the synthesized nanowires were dispersed into the ethanol. The optical transmittance and absorbance were measured with a UV-vis spectrometer (S-3100, Scinco Co.).

### Electrical characterization

For the characterization of electromechanical properties, nanowires were spin-coated on polydimethylsiloxane (PDMS, sylgard 184, Dow Corning), both ends of the nanowire films were connected to copper wires with liquid metal, and the relative resistance was measured with a source meter (Keithley 2450) at a constant voltage of 0.65 V. The nanowire films were stretched with a customized stretching machine

and their resistance changes were measured with a source meter. For the chemical stability test, each nanowire film with different coating times was exposed to $H_2O_2$ (SAMCHUN) and the relative resistance was measured with a source meter. The thickness of nanowire films was characterized with a 3D surface profilometer (Bruker, Billerica). For the stability tests in PBS (Tech & Innovation), the nanowire films were immersed in PBS (pH 7) at 37 °C for several days.

### Fabrication of the hybrid IOP sensor
Parylene C as a substrate (300 nm) (LAVIDA 110, Femto Science) was deposited on the sacrificial layer. The AZ-nLoF (Microchemicals) was spin-coated on the parylene C and patterned by photolithography for lift-off process. First, Ag@AuNW (1 mg/ml) dispersed in the mixed solution of ethanol and DI water at the volume ratio of 2:5 was spin-coated five times and annealed at 110 °C for 2 min. After annealing, the nanowire coated substrate was dipped into nitric acid (30% in DI water) to etch Ag core for 10 s. The AuHNW film was annealed at 110 °C for 2 min to fully remove the remaining nitric acid. Before D-PEDOT coating, the AuHNW film was treated with argon plasma (150 W, 2 min). For the D-PEDOT solution, D-sorbitol (20 mg/ml, Sigma Aldrich) was added into the PEDOT:PSS solution (PH1000, Clevios) with stirring for 3 h. The solution of D-PEDOT was spin-coated on the AuHNW film and annealed at 110 °C for 5 min. After that, the photoresist was removed with acetone and carefully washed with isopropyl alcohol (IPA, SAMCHUN) to prepare the hybrid IOP sensor. TPU (100 mg/ml, 1185 A, Elastollan) was finally spin-coated on the hybrid IOP sensor for passivation. The hybrid IOP sensor was transferred by dissolving PVA (363170, Sigma Aldrich) and placed into the contact lens mold. The silicone elastomer (MED-6015, Nusil) was filled and cured at 100 °C for 1 h to make soft smart contact lens. To monitor the relative resistance of IOP sensor in vitro, the copper wires were connected to the IOP sensor by silver epoxy (MED-H20E, EPO-TEK) or liquid metal (495425, Sigma Aldrich).

### Characterization of the hybrid IOP sensor
The artificial eye model was fabricated with the PDMS mixing base and a curing agent at the weight ratio of 20:1[15]. The diameter of artificial eye model was 15 mm with a thickness of 500 μm. Two scalp vein sets were inserted into the artificial eye model to control the pressure of the artificial eye model. One was connected to a customized pressure sensor and the other was connected to a syringe pump. After the smart contact lens was placed on the artificial model eye, the pressure of artificial model eye was controlled by injection or ejection of PBS. The resistance change of IOP sensor according to the pressure change was measured with a source meter at a constant voltage of 0.65 V (Supplementary Fig. 23).

### Fabrication of the flexible DDS
Parylene C (300 nm) was deposited on the sacrificial layer. Gold was deposited and patterned on the parylene C with a thickness of 100 nm for the gold channel and 500 nm for the electrode. The SU-8 2015 (Kayaku Advanced Materials, Inc) was spin-coated on the gold channel and electrode to fabricate drug reservoirs. The timolol powder (timolol maleate salt, T6394, Sigma Aldrich) was loaded into reservoirs and the PVA (100 mg/ml) solution as a protection layer was coated on the drugs and dried at room temperature. The additional parylene C layer (100 nm) was deposited to seal the DDS. The DDS was transferred by dissolving the sacrificial layer and the back side of DDS was patterned with a photoresist to selectively etch parylene C. The opened parylene C was selectively etched by RIE ($O_2$, 100 sccm, 150 W, 5 min) (Covance, Femto Science). The flexible DDS was embedded into a smart contact lens. During molding process, the gold channel and the cathode were open to allow gold dissolution by the electrochemical reaction in the tear.

### Characterization of the flexible DDS
The electrochemical reaction of gold was investigated in PBS (pH 7.4) and artificial tear with a constant voltage of 1.85 V in vitro. The voltage of 1.85 V was applied to anode and cathode of DDS for 5 min to release timolol from the reservoirs. The concentration of released timolol was quantified with a UV-vis spectrometer at the absorbance wavelength of 259 nm.

### Fabrication of highly integrated theranostic smart contact lens
For the precise integration of wireless circuits, an IOP sensor and a DDS, and all other components were sequentially fabricated on the same substrate of parylene C. First, gold with a different thickness was deposited on the substrate and patterned by photolithography for DDS electrode (100 nm) and wireless circuits (500 nm) of an antenna and a chip pad, followed by flip-chip bonding of the ASIC chip. The IOP sensor was fabricated on the same substrate by lift-off process with sequentially coating of AuHNW and D-PEDOT. After that, the parylene C was selectively etched to separate the IOP sensor and wireless circuits by RIE. SU 8 was used to pattern the drug reservoirs and the protection layer of chip, antenna and interconnection (Supplementary Fig. 24). The drug of timolol was loaded into the drug reservoirs and the protection layer of PVA was coated on the loaded drugs. TPU was spin coated on the fabricated theranostic system except drug reservoirs. Finally, parylene C (100 nm) was deposited for passivation and sealing the layer. The fabricated theranostic system was transferred by dissolving the sacrificial layer and embedded into the smart contact lens by simple molding.

### Cell viability and biosafety analysis
To assess the cell viability and biosafety of nanomaterials, fibroblast cells (NIH 3T3, mouse embryonic fibroblast) at a concentration of 5.0 × $10^3$ cells/ml were directly seeded on the parylene C as a blank substrate, AuHNW, D-PEDOT and AgNW as a control. NIH 3T3 cell line was obtained from American Type Culture Collection (CRL-1658, ATCC). The concentration of each material was same with that of IOP sensor. After cells on each film were incubated in the cell culture medium, Dulbecco's modified Eagle medium (DMEM, Thermo Fisher) for 3 days, the cells were stained with calcein AM and ethidium homodimer-1 (EthD-1) in green and red, and observed with a fluorescent optical microscope. The live/dead cell imaging kit was obtained from Thermo Fisher. The cell viability was quantified by counting live (green) and dead (red) cells on the fluorescent OM images. Smart contact lens was worn on each eye of glaucoma-induced rabbits for 1 h every other day. After 2 weeks, the corneal fluorescein staining and corneal thickness analysis were performed for the biosafety assessment.

### In vivo assessment of theranostic smart contact lens
To assess the IOP control capability of theranostic smart contact lens in vivo, the theranostic contact lenses were worn on the right (OD) and left (OS) rabbit's eyes. For the IOP monitoring, the initial IOP level was measured with a commercialized tonometer and the IOP fluctuation was monitored by wearing smart contact lens. The output codes were collected for 30 min at the interval of 5 min. After 30 min, the smart contact lens was removed from the rabbit's eye and the IOP was measured with a tonometer again. For wireless monitoring and communication, the distance between a receiver coil embedded into smart contact lens and a transmitter coil was maintained within 5 mm with the parallel alignment of the two coils. The output codes were converted into IOP (mmHg) based on our calibration standard curve for each rabbit.

### In vivo therapeutic effects of smart contact lens and eye drop
The glaucoma-induced rabbits and normal rabbits were divided into 3 groups to assess the therapeutic effects of smart contact lens and eye

drop: Glaucoma induced by MC (M0512, viscosity of 4000 cP, Sigma Aldrich) or α-chymotrypsin (C4129, Sigma Aldrich) with eye drop ($n = 3$, group 1), glaucoma induced by MC or α-chymotrypsin with smart contact lens ($n = 3$, group 2), and the normal rabbits as a control ($n = 3$, group 3). After 7 days of glaucoma induction, the treatments were performed on right eyes of rabbits and non-treatments were performed on left eyes of rabbits. The eye drop was treated on the right eyes every day for 16 days except weekends (12 days treatment in total). The timolol (1 reservoir with 38 μg) was delivered from the smart contact lens and treated on the right eyes every other day for 16 days except weekends (7 days of treatment in total).

### Histopathologic and immunohistochemical analyses

Formalin-fixed whole eyes were embedded in paraffin for the preparation of 5 μm sections. For histological evaluation, the sectioned tissues were stained with hematoxylin and eosin (H&E; ABCAM, UK) and examined under a direct light microscope. Briefly, immunohistochemical detection included antigen retrieval in 10 mM citrate buffer in a microwave oven and blocking endogenous peroxidase with 1% hydrogen peroxide. Tissues were incubated at 4 °C overnight with the following primary antibodies; GFAP (sc-51908, Santa Cruz), CD11b (ab8878, ABCAM), BDNF (ab108619, ABCAM), and Brn3a (ab345230, ABCAM). Then, the VECTASTAIN Elite ABC reagent (horse anti-mouse/rabbit IgG, Vector Laboratories, Burlingame, CA) for horseradish peroxidase was used for immunohistochemistry. After that, tissues were incubated and stained in a peroxidase substrate solution (ImmPACT DAB Substrate, Peroxidase, Vector Laboratories, Burlingame, CA) up to the desired intensity, and lightly counterstained with nuclear fast red (Abcam, UK). The dilution ratio for all antibodies listed above was 1:1000.

### Statistical analysis

All the experiments including those in Supplementary Information were carried out more than three times independently. We performed one-sided statistical analyses using a one-way analysis of variance. For all experiments, $*P < 0.05$, $**P < 0.01$, and $***P < 0.001$ were considered statistically significant. All error bars represent the standard deviation.

### Reporting summary

Further information on research design is available in the Nature Portfolio Reporting Summary linked to this article.

## Data availability

All data that support the findings of this study are available in the paper or its Supplementary Information, or from the corresponding author upon reasonable request. Source data are provided with this paper.

## Code availability

A full code is available in GitHub under the project name of "NAT_SCL_2021" (https://github.com/cheonhoo-jeon/Nat_SCL_2021) and DOI zenodo[40]. We used the commercial software programs of Xilinx ISE Design Suite (ver.14.7) and Java (ver.1.8.0_131). The LSK_v1.v was used to read the data from RF receiver and the guimake4.java was used to receive and plot the data on the computer[38]. The area fraction of each nanowire in Supplementary Fig. 2b was characterized by using the Image J program.

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

## Acknowledgements

This research was supported by the Basic Science Research Program (2020R1A2C3014070), the Korea Medical Device Development Fund grant (2020M3E5D8105732), BRIDGE Research Program (2022M3C1C3095052), and Bio & Medical Technology Development Program (2021M3E5E7021473) of the National Research Foundation (NRF) funded by the Ministry of Science and ICT, Korea.

## Author contributions

S.K.H. conceived the idea and supervised the project. T.Y.K. prepared and fabricated the samples, performed all experiments, and characterized nanomaterials. J.W.M. prepared animal models and performed immunohistochemical analyses. S.H.H., S.H.J., and H.C. performed the animal experiments and prepared samples. S.S. prepared the wireless board and integrated chip. C.-K.J. edited the manuscript. All authors contributed to writing the manuscript.

## Competing interests

The authors declare no competing interests.
