## [Peer Review File · Nature Communications]

Wireless Theranostic Smart Contact Lens for Monitoring and Control of Intraocular Pressure in GlaucomaREVIEWER COMMENTS

Reviewer #1 (Remarks to the Author):

This work reports an integrated contact lens with (1) a gold hollow nanowire based IOP sensor, showing a high strain sensitivity, chemical stability and biocompatibility, (2) a flexible DDS used for on-demand delivery of timolol for IOP control. By integrating the sensor and DDS, the IOP levels can be monitored and controlled by the smart contact lens in glaucoma induced rabbits. The hollow Au nanowires shows slightly higher transparency and strain sensitivity compared to a non-hollow version as control.

The work is interesting, but I am unclear what the advances are relative to some of the authors earlier works. The AuHNW is also not that transparent relative to existing solutions or demonstrations such as triggerfish devices.

The authors should clarify the following major comments:

- Why was a Ag@AuNW chosen as a control and not simply AuNW?
- What is the thickness of the films for characterization in Figure 2? Especially we note that the authors compared its relative resistance change with a 100 nm Au film. It would be good if the authors can also specify the thickness of the AuHNW film.
- It is important to characterize the chemical stability of the materials. Is there any reason for choosing H₂O₂ for the testing?
- Although the AuHNW is quite transparent and shows a good transmittance (Figure 2e,f), the DDS device with the ASIC chip is completely opaque. It would be good if the author can also consider the transmittance of the whole device as a benchmark.
- Figure 2g shows a very large change in resistance of over 10,000% at 30% strains. It is unclear how the devices were contacted. Also, what was the initial resistance of the film? The authors did not state whether it was a two-point or four-point probe measurement that eliminates the contact resistance effect. More details should be provided on the electromechanical characterizations.
- As a proper control, a contact lens device made using Ag@AuNW should be made and IOP relationship determined.
- There was a lack of details on the ASIC chip. While I understand the major point of this paper is the use of hollow AuNWs, more information on the ASIC should be provided, e.g., power consumption, circuit blocks, etc.
- What is the effect on the IOP for different human behaviours, for example eye blinking, closed eye, lying down etc? If they cause IOP fluctuation, is the device able to cater to different conditions?
- Figure 6b shows a relatively large spread in the values between what was measured in the contact lens vs. tonometer. Can the authors explain why? There was also less data collected at higher IOPs.
- Supp Fig 9 shows a side profile of the lens fitting. Can the authors provide top view as well? How is the perfect fitting determined?

Minor comments:

- Supplementary Figure. 3b is missing the scale bar?

Reviewer #2 (Remarks to the Author):

Integrated sensor and drug delivery system to manage glaucoma will have a high clinical impact. This study demonstrates the feasibility of their proposed integrated intraocular pressure (IOP) sensor and drug (timolol) depot that responds to changes in IOP. Though the initial feasibility is exciting, a

discussion on how the device will respond to IOP fluctuation, triggering a drug due to changes (release trigger criteria?), and potential usability are needed.

Comments:

- 1) There should be a discussion about diurnal-nocturnal IOP fluctuation. How does the proposed system determine changes and release the drug beyond the IOP fluctuation?
- 2) A discussion on how the patient will use the proposed smart contact lens is needed. Will this be worn daily? Weekly wear? (clearly define their short-term vs. long-term monitoring in terms of hours, days, weeks, months)
- 3) A discussion of why hollow nanomaterial is better (advantage) than bulk is needed.
- 4) The authors stated that 30% stretchability is reasonable. How was this determined? What would be the optimal stretchability? (similar to the current contacts lens?)
- 5) How does the device perform at a higher IOP (above 35 mm Hg)? How often would the drug release be triggered if the IOP does not respond to the treatment?
- 6) The amount of drug loaded was based on one drop of commercial eye drop. Much of the drug is lost with the eye drop before penetration into the eye. Does the contact lens provide better drug availability and facilitate diffusion into the eye? Should a lower amount be used?
- 7) Consider including the drug release complete profile with one reservoir within the main manuscript.
- 8) It is not clear what Figure 4G demonstrates. Is this a cumulative amount of drug release?
- 9) Clearly state how long the animals had contact lenses on for the biosafety study.
- 10) Figure 6 should show the IOP readings on Day 2 and Day 4. Does the treatment provide a relatively sustained decrease in IOP these days? How fast does IOP increase as the drug therapeutic is decreasing? What was the rationale for treating every other day with the device?

Reviewer #3 (Remarks to the Author):

General comments

The idea to develop a theranostic contact lens for monitoring and treating intraocular pressure of glaucoma patients has been a goal of several laboratories. The proposed approach is quite interesting and novel. However, it difficult to critically evaluate as many of the methods are poorly described and some even seem incomplete.

It appears that there are drug reservoirs that can be released on demand or in response to increased eye pressure.

1. With only a few reservoirs, the lens would need to be replaced regularly. For a once daily drug, this might happen at just a few day intervals. For a twice daily drug, it might be every 2 days. For a drug administered more frequently, it might be daily. This could be a major impediment to its use. Would such an approach with this lens be safe? (eg risk to corneal epithelium and keratitis, ulcer, etc) How can this be addressed?
2. The drug is released over ~30 min. How does this influence efficacy and safety compared to an eye drop that is released as a bolus? Bathing tissue with a drug is very different from brief exposure. The efficacy studies are performed with two poorly reproducible models (alpha-chymo and methycellulose). It is unclear why you employed these models and evaluated so few rabbits?

Specific comments

Minor comments

1. L 15 –Although glaucoma is irreversible, it is not “menacing” as most affected patients do not lose useful vision.
2. L30 IOP does not indicate glaucoma condition as majority of individuals with high iOP do not have glaucoma and as many as one half of patients with glaucoma do not have high IOP.
3. L34 Tonometry is typically not uncomfortable. Tonometers are not typically thought to be expensive. With current instruments, clinical training needed is minimal and the measurements are

easy to perform with just a few minutes of training.

4. L36 Unclear what is meant by “critical mismatch” –IOP varies throughout the day and also day to day.

5. L38 Adherence is a problem with eye drop treatment, however the text is inaccurate.

6. L140 Unclear how it is tested to measure resistance with change in IOP

Responses to Reviewer's Comments

Thank you so much for your kind review on our manuscript. Reflecting your comments, this manuscript has been revised as described in the following responses.

Reviewer #1 (Remarks to the Author):

This work reports an integrated contact lens with (1) a gold hollow nanowire based IOP sensor, showing a high strain sensitivity, chemical stability and biocompatibility, (2) a flexible DDS used for on-demand delivery of timolol for IOP control. By integrating the sensor and DDS, the IOP levels can be monitored and controlled by the smart contact lens in glaucoma induced rabbits. The hollow Au nanowires shows slightly higher transparency and strain sensitivity compared to a non-hollow version as control.

→ We sincerely appreciate your valuable review and comments.

The work is interesting, but I am unclear what the advances are relative to some of the authors earlier works. The AuHNW is also not that transparent relative to existing solutions or demonstrations such as triggerfish devices.

→ Thank you so much for your valuable comments. Although many researchers have studied smart contact lenses for monitoring or control of IOP, there was no report on the theranostic smart contact lens which can simultaneously monitor and control IOP in response to the sensing results. Our theranostic smart contact lens was fully integrated with an IOP sensor, a DDS and wireless circuits for the feedback system. In this regard, our theranostic smart contact lens is a more advanced system than the traditional smart contact lens such as Triggerfish device. Furthermore, AuHNW was developed for the advanced IOP sensor with high stability, sensitivity and biocompatibility in addition to the transparency. The AuHNW showed many advantages compared to non-hollow nanowires and silver nanowires. Taken together, we can clearly insist the novelty of our theranostic smart contact lens with the AuHNW IOP sensor. For better understanding, we discussed the novelty of our theranostic smart contact lens with **Supplementary Table 1** in the revised manuscript (**Supplementary Information, page 2, line 17**).

The authors should clarify the following major comments:

1. Why was a Ag@AuNW chosen as a control and not simply AuNW?

→ Many thanks for your valuable comments. Because AuHNW was prepared by using the Ag@AuNW, we used the Ag@AuNW as a control. For comparison, we added the size distribution of Ag@AuNW and AuHNW. There was no significant difference in length. In addition, because it's difficult to prepare the similar length and distribution of AuNWs with those of AuHNWs, we did not use AuNW as a control (page 13, line 290 and Supplementary Fig. 19).

2. What is the thickness of the films for characterization in Figure 2? Especially we note that the authors compared its relative resistance change with a 100 nm Au film. It would be good if the authors can also specify the thickness of the AuHNW film.

→ Thank you so much for your valuable comments and suggestion. The thickness of AuHNW films in Fig. 2 was ca. 100 nm. In addition, there was no significant difference in the relative resistance change with increasing thickness of AuHNW as shown in Fig. 2i.

3. It is important to characterize the chemical stability of the materials. Is there any reason for choosing H₂O₂ for the testing?

→ Many thanks for your valuable comments. As you know, H₂O₂ is one of the strong oxidizing agents in the body and used for the chemical stability tests. In addition, we also checked the stability of AuHNW in PBS (37 °C) for 2 weeks to confirm the stability in the physiological condition (Supplementary Fig. 4d).

4. Although the AuHWN is quite transparent and shows a good transmittance (Figure 2e,f), the DDS device with the ASIC chip is completely opaque. It would be good if the author can also consider the transmittance of the whole device as a benchmark.

→ We appreciate your valuable comments. As you mentioned, the DDS device with the ASIC chip is inevitably opaque. We will consider minimizing the ASIC chip size and using transparent drug delivery hydrogels for more futuristic smart contact lens. Anyway, our smart contact lens containing AuHNW IOP sensor is meaningful with high sensitivity, stability, biocompatibility and transparency. As recommended, we assessed the transmittance of the whole device lens as a benchmark (Supplementary Fig. 5b).

5. Figure 2g shows a very large change in resistance of over 10,000% at 30% strains. It is unclear how the devices were contacted. Also, what was the initial resistance of the film? The authors did not state whether it was a two-point or four-point probe measurement that eliminates the contact resistance effect. More details should be provided on the electromechanical characterizations.

→ Many thanks for your valuable comments. The initial resistances were about 26 Ω and 196 Ω for Ag@AuNW and AuHNW, respectively. As shown in Supplementary Fig. 5a, because the core of Ag was etched, the sheet resistance of AuHNW was slightly increased compared to that of Ag@AuNW. The electromechanical properties were analyzed by two-point probe measurement. The detailed experimental methods were added in the revised manuscript (page 16, line 341). In brief, each nanowire was coated on the PDMS film and the end of nanowire film was connected to Cu wire with liquid metal. After that, the nanowire coated PDMS was stretched with a customized stretcher and the relative resistance was measured with a source meter (Keithley).

6. As a proper control, a contact lens device made using Ag@AuNW should be made and IOP relationship determined.

→ Thank you so much for your valuable comments. As recommended, we performed additional experiments with Ag@AuNW and AuHNW IOP sensors. The AuHNW IOP sensor showed higher sensitivity in response to IOP than that of Ag@AuNW IOP sensor. Furthermore, AuHNW IOP sensor could monitor a wide range of IOP over 50 mmHg (page 7, line 151 and Supplementary Fig. 6d, e).

7. There was a lack of details on the ASIC chip. While I understand the major point of this paper is the use of hollow AuNWs, more information on the ASIC should be provided, e.g., power consumption, circuit blocks, etc.

→ Thank you so much for your valuable comments. As you recommended, we described detailed information on our ASIC chip (page 11, line 229) with our relevant references for better understanding in the revised manuscript (Supplementary Fig. 12).

8. What is the effect on the IOP for different human behaviours, for example eye blinking, closed eye, lying down etc? If they cause IOP fluctuation, is the device able to cater to different conditions?

→ We appreciate your valuable comments. As you mentioned, IOP can be changed by different human behaviors such as eye blinking, closed eye, and lying down. *In vivo* experiments, however, were performed with excluding other factors such as posture changes and eye blinking by fixing the rabbits or under anesthesia for the accurate characterization of IOP sensors in the therapeutic smart contact lens. In the proposed device, there was no program or algorithm to exclude the unusual IOP fluctuation. For further commercial development, we will adopt such a program to exclude the unusual IOP fluctuation. We discussed the IOP fluctuation issue in the revised manuscript (Supplementary Information page 2, line 32)

9. Figure 6b shows a relatively large spread in the values between what was measured in the contact lens vs. tonometer. Can the authors explain why? There was also less data collected at higher IOPs.

→ Many thanks for your valuable comments. The mechanism of smart contact lens is different from that of tonometer for the IOP measurement. However, as shown in Supplementary Fig. 14, there was no significant difference in the IOP changes measured by the smart contact lens and the tonometer. The maximum IOP difference by the two devices was less than 3.5 mmHg. Furthermore, as shown in Fig. 6b, there was a strong correlation between measured IOPs by tonometer and smart contact lens ($R^2 = 0.94$). These results clearly indicate that our smart contact lens can accurately monitor IOP in correlation with that by the commercial tonometer. As recommended, we carried out further experiments and added more data at higher IOP levels (Fig. 6b).

10. Supp Fig 9 shows a side profile of the lens fitting. Can the authors provide top view as well? How is the perfect fitting determined?

→ Thank you so much for your valuable comments. We previously added the top view for the perfect fitting of smart contact lens on the rabbit eye (Fig. 6a). If the smart contact lens is bigger than the eye, the smart contact lens can be easily removed from the eye and there is a gap between contact lens and eye. Thus, we designed the smart contact lens with a diameter of 7.5 mm for perfect fitting without gap between the smart contact lens and rabbit eyes, as shown in the side view of smart contact lens on the rabbit eye (Supplementary Fig. 9).

Minor comments: Supplementary Figure. 3b is missing the scale bar?

→ We are sorry for our mistake. We added the scale bar in the SEM (Supplementary Fig. 3).

Reviewer #2 (Remarks to the Author):

Integrated sensor and drug delivery system to manage glaucoma will have a high clinical impact. This study demonstrates the feasibility of their proposed integrated intraocular pressure (IOP) sensor and drug (timolol) depot that responds to changes in IOP. Though the initial feasibility is exciting, a discussion on how the device will respond to IOP fluctuation, triggering a drug due to changes (release trigger criteria?), and potential usability are needed.

→ We sincerely appreciate your valuable review and comments.

Comments:

1) There should be a discussion about diurnal-nocturnal IOP fluctuation. How does the proposed system determine changes and release the drug beyond the IOP fluctuation?

→ Thank you so much for your valuable comments. As recommended, we discussed the diurnal-nocturnal IOP fluctuation in the revised manuscript (Supplementary Information page 2, line 32). *In vivo* experiments were performed with excluding other factors such as posture changes and eye blinking by fixing the rabbits or under anesthesia for the accurate characterization of IOP sensors in the theranostic smart contact lens. As a result, we could confirm the efficient IOP reduction by just releasing drugs from the theranostic smart contact lens. For further commercial development, we will adopt an algorithm or a program to exclude the unusual IOP fluctuation.

2) A discussion on how the patient will use the proposed smart contact lens is needed. Will this be worn daily? Weekly wear? (clearly define their short-term vs. long-term monitoring in terms of hours, days, weeks, months)

→ Many thanks for your valuable comments. Currently, there are commercially available silicone contact lenses which can be worn for a month or longer. According to the amount of drug reservoirs, our smart silicone contact lens can be used for a day, three days, and a week. Our theranostic contact lens would be greatly beneficial for the automatic control of IOP by releasing the drug in response to the increased IOP. As recommended, we discussed this issue in the revised manuscript (Supplementary Information page 3, line 50).

3) A discussion of why hollow nanomaterial is better (advantage) than bulk is needed.

→ Thank you so much for your valuable comments. As recommended, we discussed the novelty of AuHNW in the revised manuscript (Supplementary Information page 2, line 17). In brief, AuHNW has high sensitivity to IOP, high stability to other external stimulations such as chemical oxidization and temperature, high biocompatibility and high optical transmittance.

4) The authors stated that 30% stretchability is reasonable. How was this determined? What would be the optimal stretchability? (similar to the current contacts lens?)

→ Many thanks for your valuable comments. As reported elsewhere (*Nat. Commun.* 8, 1-8, 2017 / *Sci. Adv.* 4, eaap9841, 2018 / *Nat. Commun.* 12, 1-11, 2021), the maximum strain is generally regarded to be less than 30 % for the use of contact lens. In addition, 1 mmHg increase of IOP corresponds to only 0.03 % strain. Thus, we assumed that the necessary stretchability of IOP sensor would be less than 30 % for the general application.

5) How does the device perform at a higher IOP (above 35 mm Hg)? How often would the drug release be triggered if the IOP does not respond to the treatment?

→ Thank you so much for your valuable comments. We conducted additional experiments to assess the IOP sensing capability of smart contact lens in a wide range. Our theranostic smart contact lens appeared to monitor the IOP from 0 mmHg to 200 mmHg. The AuHNW IOP sensor showed higher sensitivity than the Ag@AuNW IOP sensor (Supplementary Fig. 6d, e). Furthermore, we confirmed that our DDS could effectively reduce the IOP in response to the IOP sensing. We will investigate various unusual situations later for further development.

6) The amount of drug loaded was based on one drop of commercial eye drop. Much of the drug is lost with the eye drop before penetration into the eye. Does the contact lens provide better drug availability and facilitate diffusion into the eye? Should a lower amount be used?

→ Thank you so much for your valuable comments. As you mentioned, we determined the amount of loaded timolol in a reservoir of smart contact lens as one drop of commercial eye drop for the direct comparison of therapeutic effects between eye drop and smart contact lens. The drug eluting contact lens can generally enhance the bioavailability of drugs by inhibiting the wash out of drugs from repeated eye blinking and tears (*J. Control. Release* 281, 97-118, 2018 / *Molecules* 26, 5577, 2021). As reported elsewhere (*J. Control. Release* 226, 47-56, 2016), a low amount of drugs from

the hydrogel contact lens could effectively reduce the IOP level. Thus, we believe that a lower amount of timolol can efficiently reduce the IOP. The exact amount of drugs would be determined on the basis of further clinical studies. Our theranostic smart contact lens for simultaneous monitoring and control of IOP would contribute to open a new paradigm of glaucoma management.

7) Consider including the drug release complete profile with one reservoir within the main manuscript.

→ We would like to thank for your valuable comments. As recommended, we included the drug release profile in the revised manuscript (Fig. 4g).

8) It is not clear what Figure 4G demonstrates. Is this a cumulative amount of drug release?

→ Thank you so much for your valuable comments. Fig. 4G shows the timolol release profile from a reservoir and Fig. 4H does the cumulative release of timolol by sequentially operating three different reservoirs in the DDS of smart contact lens in the revised manuscript. From the results, we could confirm that our DDS system can be used for on-demand timolol release for the feedback system (Fig. 4h and page 9, line 193).

9) Clearly state how long the animals had contact lenses on for the biosafety study.

→ Many thanks for your valuable comments. The biosafety was investigated by wearing contact lens on the eye for 1 h every other day for 2 weeks (page 19, line 420).

10) Figure 6 should show the IOP readings on Day 2 and Day 4. Does the treatment provide a relatively sustained decrease in IOP these days? How fast does IOP increase as the drug therapeutic is decreasing? What was the rationale for treating every other day with the device?

→ Thank you so much for your valuable comments. We added the IOP levels of Day 2 and Day 4 in the revised manuscript (Supplementary Information Fig. 15a). We performed additional experiments to study the reducing profile of IOP by the treatment with smart contact lens. Upon the drug treatment with smart contact lens, the IOP was decreased quickly, maintained within the reduced IOP for 18 h, and returned to the initial IOP level after 24 h (Supplementary Information Fig. 15b). From the ethical issues for animal experiments with anesthesia, we decided to treat every other day with the device.

Reviewer #3 (Remarks to the Author):

General comments: The idea to develop a theranostic contact lens for monitoring and treating intraocular pressure of glaucoma patients has been a goal of several laboratories. The proposed approach is quite interesting and novel. However, it difficult to critically evaluate as many of the methods are poorly described and some even seem incomplete. It appears that there are drug reservoirs that can be released on demand or in response to increased eye pressure.

→ We sincerely appreciate your valuable review and comments.

1. With only a few reservoirs, the lens would need to be replaced regularly. For a once daily drug, this might happen at just a few day intervals. For a twice daily drug, it might be every 2 days. For a drug administered more frequently, it might be daily. This could be a major impediment to its use. Would such an approach with this lens be safe? (eg risk to corneal epithelium and keratitis, ulcer, etc) How can this be addressed?

→ Many thanks for your valuable comments. Our theranostic smart contact lens is comprised of six drug reservoirs as a prototype for simultaneous monitoring and control of IOP. We believe that our prototype smart contact lens can be used for timolol delivery once a day for 6 days on the basis of our results (Fig. 4 and 6). Furthermore, our smart contact lens can be used for more than 6 days by increasing the number of drug reservoirs with high potency drugs. We have shown the biosafety of our theranostic smart contact lens in Figure 5. There were no significant damage and inflammation in the cornea without the corneal thickness change.

2. The drug is released over ~30 min. How does this influence efficacy and safety compared to an eye drop that is released as a bolus? Bathing tissue with a drug is very different from brief exposure. The efficacy studies are performed with two poorly reproducible models (alpha-chymo and methycellulose). It is unclear why you employed these models and evaluated so few rabbits?

→ Thank you so much for your valuable comments. We additionally studied the reducing profile of IOP by the treatment with smart contact lens. Upon the release of timolol from smart contact lens, the IOP was decreased quickly, maintained within the reduced IOP for 18 h, and returned to the initial IOP level after 24 h (Supplementary Information Fig. 15b). As reported elsewhere (*J.*

Control. Release 305, 18-28, 2019 / *J. Control. Release* 226, 47-56, 2016), a very small amount (lower than 200 ng) of timolol appeared to successfully reduce the IOP.

We used alpha-chymo and methylcellulose models because of their similar characteristics with real glaucoma. Many previous studies have used these models to investigate the glaucoma treatments (*J. Ocul. Pharmacol. Ther.* 5, 93-98, 1989 / *Nanoscale* 9, 11754-11764, 2017 / *Int. J. Mol. Sci.* 21, 9267, 2020). The retinal ganglion cell densities of these groups were slightly lower than normal rabbits, which is the very characteristics of real glaucoma. The high IOP level was maintained for more than 2 weeks in the glaucoma models (more than 4 weeks in some cases).

Specific comments

Minor comments

1. L 15 – Although glaucoma is irreversible, it is not “menacing” as most affected patients do not lose useful vision.

→ Many thanks for your valuable comments. To avoid misunderstanding, we deleted the word of “menacing” in the revised manuscript (page 2, line 15).

2. L30 IOP does not indicate glaucoma condition as majority of individuals with high IOP do not have glaucoma and as many as one half of patients with glaucoma do not have high IOP.

→ Thank you so much for your valuable comment. We agree with your comments. However, IOP is one of the most important indicators for both patients of normal tension glaucoma and general glaucoma. The majority of patients are recommended for monitoring IOP and reducing IOP to the lower range in both cases of glaucoma. To avoid misunderstanding, we revised the manuscript correctly (page 2, line 30) with discussion (Supplementary Information page 3, line 41).

3. L34 Tonometry is typically not uncomfortable. Tonometers are not typically thought to be expensive. With current instruments, clinical training needed is minimal and the measurements are easy to perform with just a few minutes of training.

→ Many thanks for your valuable comments. We agree with your comments, but tonometers cannot be used for the continuous monitoring of IOP. To avoid misunderstanding, we revised the manuscript correctly (page 2, line 35).

4. L36 Unclear what is meant by “critical mismatch” – IOP varies throughout the day and also day to day.

→ Thank you so much for your valuable comments. To avoid misunderstanding, we revised the manuscript correctly (page 3, line 38).

5. L38 Adherence is a problem with eye drop treatment, however the text is inaccurate.

→ Many thanks for your valuable comments. As recommended, we revised our manuscript to describe the challenges of eye drop treatment in glaucoma patients more accurately (page 3, line 42).

6. L140 Unclear how it is tested to measure resistance with change in IOP

→ Thank you so much for your valuable comments. We previously described the measurement method of resistance in response to IOP. For easy understanding, we added a photo-image of in vitro experiments set up (Supplementary Fig. 22).

REVIEWER COMMENTS

Reviewer #1 (Remarks to the Author):

The authors addressed most of my concerns with the manuscript.
Some questions remain:

1. Some details are still missing. What was the ASIC process used? Which technology node was it? How thick was the ASIC chip? Most ASIC process do not result in flexible chips.
2. Liquid metal was used for the connection between the patterned AuHNW and external wire for electrical test in vitro. Fig. 3c show the IOP and wireless circuit. What is used for the connection between the hybrid IOP sensor and electrode to form a stable connection and to ensure the test signal in vivo all coming from the hybrid structure? A degradation at the connection part may cause irreversible increasing resistance which lead to a misleading DDS.
3. L149 show the reliability, only the first three measurements with loading curve are shown in Fig. 3f. What are the first, second and third measurements? Three cyclic tests? The result is not convincing that the sensor exhibits a good reliability if only based on three repeated tests.
4. Why is there only stretchability test for pure AuHNWs films, and no stretchability test for AuHNW/D-PEDOT as the real IOP pressure sensing elements.
5. The authors mentioned: "Because glaucoma patients have different drug responses in certain physiological conditions, this feedback system for both monitoring and control of IOP can maximize the compliance of glaucoma patients with the minimized side effect." Please provide studies that support this statement. What are the physiologic conditions?

Issues with figures:

1. Lack of scale bars in Fig. 2e

Reviewer #2 (Remarks to the Author):

The authors addressed concerns in the revised manuscript.

Reviewer #3 (Remarks to the Author):

The authors have responded to each of my comments, however I remain unconvinced about the safety and efficacy.

I do question the safety of this device when used in human patients. The Triggerfish lens, to which the authors refer and compare, was FDA approved only for 24 hour wear. Despite FDA approval, it was never marketed in US. In Europe, it was not been widely used due to safety and efficacy.

There also is little information provided that would enable the reader to ascertain whether human vision would be reduced by the lens and the drug release. "Marinating" a cornea with a drug is very different from a bolus delivered with an eyedrop. It is inconceivable that a human cornea would tolerate repeated and lengthy drug exposures. This is especially the case for timolol, a drug with many known ocular side effects after topical drug delivery (including anesthesia, keratitis, etc).

Responses to Reviewer's Comments

Thank you so much for your kind review on our manuscript. Reflecting your comments, this manuscript has been revised as described in the following responses.

Reviewer #1 (Remarks to the Author): The authors addressed most of my concerns with the manuscript. Some questions remain:

→ We sincerely appreciate your valuable review and comments.

1. Some details are still missing. What was the ASIC process used? Which technology node was it? How thick was the ASIC chip? Most ASIC process do not result in flexible chips.

→ Thank you so much for your valuable comments. We described the details of ASIC chip in our previous report (*IEEE J. Solid-State Circuits* 55, 856-867, 2020 and *Science Advances* 6, eaba3252, 2020). The more information on the ASIC chip was described in the revised manuscript (Supplementary Information, page 8, line 154). In brief, the ASIC chip was fabricated using a 180 nm complementary metal-oxide semiconductor (CMOS) process in a die area of 2.25 mm² (Supplementary Fig. 12). The thickness of ASIC chip was ca. 200 μm. Currently, we reduced the thickness of ASIC chip to 60 μm by using the chip polishing technique. As you mentioned, because ASIC chip was rigid, we introduced a protection layer of SU8 on the ASIC chip for their stability (Supplementary Fig. 24).

2. Liquid metal was used for the connection between the patterned AuHNW and external wire for electrical test in vitro. Fig. 3c show the IOP and wireless circuit. What is used for the connection between the hybrid IOP sensor and electrode to form a stable connection and to ensure the test signal in vivo all coming from the hybrid structure? A degradation at the connection part may cause irreversible increasing resistance which lead to a misleading DDS.

→ Many thanks for your valuable comments. Liquid metal or silver epoxy was used to connect the external wire to the gold electrode and hybrid IOP sensor for in vitro tests. However, we did not use the liquid metal or silver epoxy to interconnect the gold electrode and hybrid IOP sensor for wireless communication. The hybrid IOP sensor was directly coated and patterned on the gold electrode. As you mentioned, the degradation at the connection part can cause irreversible

increasing resistance. To solve this problem, we introduced a protection layer of SU8 at the interconnection part of hybrid IOP sensor and electrode for wireless communication (page 18, line 409 and Supplementary Fig. 24).

3. L149 show the reliability, only the first three measurements with loading curve are shown in Fig. 3f. What are the first, second and third measurements? Three cyclic tests? The result is not convincing that the sensor exhibits a good reliability if only based on three repeated tests.

→ Many thanks for your valuable comments. In our previous manuscript, we demonstrated the reliability of our IOP sensor with the repeated measurement data (Fig. 3f and Supplementary Fig. 6), correlation between IOP sensor and tonometer (Fig. 6b), and the Bland-Altman plot (Supplementary Fig. 14). In response to your comments, however, we additionally performed five repeated tests with a single IOP sensor and confirmed the reliability of our IOP sensor (Fig. 3f).

4. Why is there only stretchability test for pure AuHNWs films, and no stretchability test for AuHNW/D-PEDOT as the real IOP pressure sensing elements.

→ We appreciate your valuable comments. Because the sensitivity of each nanowire is depending on the structure, we directly compared the sensitivity of Ag@AuNW and AuHNW. In addition, as you mentioned, we also assessed the sensitivity of AuHNW/D-PEDOT and Ag@AuNW/D-PEDOT (Supplementary Fig. 3). The sensitivity of AuHNW/D-PEDOT was higher than that of Ag@AuNW/D-PEDOT.

5. The authors mentioned: "Because glaucoma patients have different drug responses in certain physiological conditions, this feedback system for both monitoring and control of IOP can maximize the compliance of glaucoma patients with the minimized side effect." Please provide studies that support this statement. What are the physiologic conditions?

→ Thank you so much for your valuable comments. Because each glaucoma patient has different glaucoma progression, target intraocular pressure (IOP) is generally different for each glaucoma patient as reported elsewhere (*Surv. Ophthalmol.* 47, S77-S89, 2002 / *Indian J. Ophthalmol.* 66, 495-505, 2018). Furthermore, the side effect and IOP reduction efficiency are different for each glaucoma patient (*Drugs* 17, 38-55, 1979). The adherence of glaucoma patients is also important for glaucoma treatments (*Am. J. Ophthalmol.* 140, 598, 2005). Accordingly, it is highly important to have the optimal therapeutic plan for each glaucoma patient. Our theranostic smart contact lens

can continuously monitor IOP and deliver appropriate drugs in consideration of patient's condition. This system can maximize the therapeutic effect and minimize the side effect with a personalized treatment plan for futuristic glaucoma treatment. We deleted the word 'physiological' to avoid misunderstanding in the revised manuscript (page 14, line 305 and Supplementary Information, page 3, line 57).

Issues with figures:

1. Lack of scale bars in Fig. 2e

→ Many thanks for your valuable comments. We added the scale bar in the OM image (Fig. 2e).

Reviewer #2 (Remarks to the Author): The authors addressed concerns in the revised manuscript.

→ We sincerely appreciate your positive and encouraging review comments.

Reviewer #3 (Remarks to the Author): The authors have responded to each of my comments, however I remain unconvinced about the safety and efficacy.

→ We sincerely appreciate your valuable review and comments.

I do question the safety of this device when used in human patients. The Triggerfish lens, to which the authors refer and compare, was FDA approved only for 24 hour wear. Despite FDA approval, it was never marketed in US. In Europe, it was not been widely used due to safety and efficacy.

→ Many thanks for your valuable comments. As you mentioned, despite the FDA approval of Triggerfish, it showed low sensitivity with the inconvenient accessories, making difficult the commercial application. In contrast, we greatly improved the sensitivity of IOP sensor and the performance of our smart contact lens for further commercial application with a thin ASIC chip. On the basis of the results in this work, we will carry out the clinical trial of our device in near future. Our smart contact lens would be successfully commercialized with better sensitivity, safety and convenience for the patients with highly biocompatible nanomaterials, new lens materials and the innovative chip design. We discussed this issue for further clinical development and commercialization in the revised manuscript (Supplementary Information, page 2, line 31).

There also is little information provided that would enable the reader to ascertain whether human vision would be reduced by the lens and the drug release. "Marinating" a cornea with a drug is very different from a bolus delivered with an eyedrop. It is inconceivable that a human cornea would tolerate repeated and lengthy drug exposures. This is especially the case for timolol, a drug with many known ocular side effects after topical drug delivery (including anesthesia, keratitis, etc).

→ Many thanks for your valuable comments. The drug delivery from our smart contact lens is not different from that by the eyedrop. The drug is only released by the electrical signal just like the eyedrop. As you mentioned, it is critical to reduce the side effect of drugs for glaucoma treatment. In addition, the target IOP can be different for each patient in the different condition of glaucoma. Accordingly, the exact dosage and therapeutic plan should be determined by the discussion with ophthalmologists. Our smart contact lens would be greatly helpful for the ophthalmologists to determine the therapeutic plan by analyzing the individual data with long-term and continuous IOP monitoring. Furthermore, although we used the same amount of timolol (50 µg) for the direct comparison of therapeutic effects between eye drop and smart contact lens, the loaded amount and type of drugs can be optimized by the therapeutic plans of ophthalmologists. On top of that, our smart contact lens with a feedback system can continuously monitor the IOP and deliver the appropriate amount of drugs in response to patients' IOP conditions. Thus, it would maximize the therapeutic effect and minimize the side effect of drugs by avoiding the unnecessary treatment. In our clinical tests, we will confirm the safety of drugs repeatedly released from our smart contact lens (Supplementary Information, page 3, line 54).